# OCTrack: Benchmarking the Open-Corpus Multi-Object Tracking

**Zekun Qian**[1], **Ruize Han**[2,3], **Wei Feng**[1], **Junhui Hou**[2], **Linqi Song**[2], **Song Wang**[4]

[1]Tianjin University, [2]Shenzhen Institution of Advanced Technology,
[3]City University of Hong Kong, [4]University of South Carolina
`rz.han@cityu.edu.hk`

## Abstract

We study a novel yet practical problem of open-corpus multi-object tracking (OCMOT), which extends the MOT into localizing, associating, and recognizing generic-category objects of both seen (base) and unseen (novel) classes, but without the category text list as prompt. To study this problem, the top priority is to build a benchmark. In this work, we build OCTrackB, a large-scale and comprehensive benchmark, to provide a standard evaluation platform for the OCMOT problem. Compared to previous datasets, OCTrackB has more abundant and balanced base/novel classes and the corresponding samples for evaluation with less bias. We also propose a new multi-granularity recognition metric to better evaluate the generative object recognition in OCMOT. By conducting the extensive benchmark evaluation, we report and analyze the results of various state-of-the-art methods, which demonstrate the rationale of OCMOT, as well as the usefulness and advantages of OCTrackB.

## 1 Introduction

Multi-object tracking (MOT), which involves detecting and associating the targets of interest in a video, is a classical and fundamental problem with many real-world applications, such as video surveillance, autonomous driving, *etc*. Recently, MOT has attracted broad attention with numerous algorithms and datasets [1, 2, 3, 4, 5, 6]. For many years, MOT has mainly focused on the target of humans, *e.g.*, the datasets of MOT15 [7], MOT20 [8], DanceTrack [9]. Several works also focus on traffic scenes and aim to track vehicles, such as the well-known KITTI [10] dataset.

In real-world scenes, the categories in videos are diverse, far from being limited to humans and vehicles. TAO [11], as the first work, constructs a large-scale benchmark to study tracking any category of target, with a total of 833 object classes. During the same period, GMOT-40 [12] builds a generic multi-object tracking benchmark with 10 object classes but more dense objects per frame. With the number of categories increasing in the MOT task, the evaluation metrics evolve from just object localization and association to also include class recognition. A new metric TETA (tracking-every-thing accuracy) is proposed [13] to evaluate the generic MOT from the above three aspects. More recently, open-world MOT (OWMOT) [14] is proposed to train a tracker using the samples from 'base classes', and test it on videos containing objects from 'novel classes'. The tracker must recognize the base-class objects and identify all other unseen classes as 'new'. Further, open-vocabulary MOT (OVMOT) [15] aims to not only distinguish the novel-category objects but also classify each object, typically achieved by a pre-trained multi-modal model, *e.g.*, CLIP [16].

Undoubtedly, the development of MOT from specific-category to generic-category and further to open-world/vocabulary settings is becoming increasingly practical. A remaining problem in the latest OVMOT is that, during testing, a predefined category list of base and novel classes is required as the text prompts for the classification task, as shown in Figure 1(a). However, obtaining this list in real

applications is not easy, especially for novel classes, which are termed novel because the categories are previously unknown. This way, in this work, we propose a new problem called Open-Corpus Multi-Object Tracking (OCMOT), which treats the object recognition task as a generative problem, rather than the classification problem in OVMOT, as shown in Figure 1(b), where the category list is no longer required.

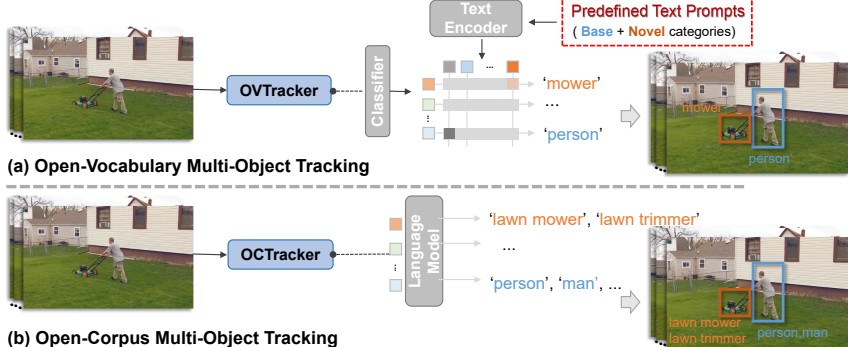

Figure 1: Illustration of the open-vocabulary and open-corpus multi-object tracking.

To study OCMOT, the top priority is to build a benchmark. Previous work OVTrack [15], directly uses TAO's validation and test sets by maintaining the classes overlapped with LVIS [17] for data selection, to construct the OVMOT evaluation datasets. Such simple category intersection operation significantly decreases the number of classes and testing samples. In this work, we build a new and comprehensive evaluation benchmark, OCTrackB, following the principles of category enrichment, sample enrichment, and semantic compatibility. Compared to previous datasets, OCTrackB offers more diverse and balanced base/novel classes, along with abundant videos for evaluation with less bias. In summary, the main contributions of this paper include:

• We propose a new problem open-corpus multi-object tracking OCMOT, which relaxes the restriction in open-vocabulary tracking by no longer requiring the given class list. OCMOT further releases the potential of MOT for practical applications in open scenes.

• We build OCTrackB, a large-scale and comprehensive benchmark, to provide the standard evaluation platform for the OCMOT problem. We also propose a multi-granularity recognition metric to further improve the performance evaluation.

• We develop the first baseline method for OCMOT. On OCTrackB, we conduct benchmark evaluation experiments and report the results of our baseline and other state-of-the-art comparison methods. Experimental results demonstrate the rationale of the OCMOT problem and the usefulness and advantages of OCTrackB.

## 2   Related Work

**Multiple Object Tracking (MOT).** The dominant approach in MOT is the tracking-by-detection framework [18], which initially identifies objects in each frame and then associates them across frames using various cues such as object appearance features [19, 20, 21, 22, 23, 24, 4, 25], 2D motion features [26, 27, 28, 29, 30], or 3D motion features [31, 32, 33, 34, 35, 36]. Some approaches enhance tracking performance by leveraging graph neural networks [37, 38] or transformers [5, 39, 6, 40] to learn association relationships among the objects of different frames. To extend the object categories in the MOT task, the TAO benchmark [11] has been proposed, which handles the MOT under various object categories with a long-tail distribution. Several follow-up works are proposed to evaluate this benchmark including AOA [41], GTR [40], TET [42], QDTrack [20], *etc.* Although these methods perform effectively, they are confined to closed-set object categories, *i.e.*, the object categories in training and testing sets are overlapped. This is unsuitable for diverse open-world scenarios with new categories. Differently, this work tracks objects of categories whether or not appearing during training, and generates their classes, which significantly expands the practical application for tracking.

**Open-World MOT** has not been extensively explored. Some existing related works [43, 44] adopt the class-agnostic detectors with general trackers to implement open-world tracking. These methods focus solely on tracking salient objects in the scene without considering specific categories. The recent TAO-OW [45] takes a step further by considering the challenges of classification in open-world tracking, dividing all objects into known and unknown categories. In this work, category-aware

open-world tracking is achieved by tracking objects of both known and unknown categories. While this advancement is a step forward in open-world tracking, it still falls short in the recognition of specific object classes in unknown categories. Further, OVTrack [15] incorporates open vocabulary into the tracking task as OVMOT, providing a baseline method and benchmark built upon the TAO dataset. Although it is much more practical, a remaining problem is the requirement for the predefined category list during the testing stage. Differently, our OCMOT does not require predefined category names as in the OVMOT task. Instead, it directly generates target category names using a generative model, which overcomes the limitations of the OVMOT problem and enhances generalizability.

**MOT Benchmarks.** Benchmarks have been pivotal in advancing the development of MOT. Early datasets like PETS2009 [46] focused on pedestrian tracking with limited video sequences. The MOT Challenge [7, 8] introduced more crowded scenes, significantly progressing the field. KITTI [10] and BDD100K [47], designed for autonomous driving, focus on tracking vehicles and pedestrians. Specialized datasets such as DanceTrack [9], SportsMOT [48], and AnimalTrack [49] handle specific scenarios like dancing, sports, and wildlife. UAVDT [50] and VisDrone [51] support aerial tracking. Despite these advancements, many benchmarks have limited object categories. Recent video datasets like GMOT-40 [12] and YT-VIS [52] aim to address specific tasks like one-shot MOT and video instance segmentation but still fall short in supporting a wide range of categories. A large-scale dataset TAO [11] annotates 833 categories, offering a broader platform for studying object tracking on long-tailed distributions. Based on TAO, OVTrack [15] builds the OVTAO evaluation datasets. Since the current popular open-vocabulary related tasks commonly use the LVIS [17] dataset for base/novel category splits, OVTrack also follows this setting. However, the proportion of novel classes in OVTAO accounts for only 10% of the original novel classes in LVIS, with around 30 classes. The limited classes hinder the effective validation of the algorithm's performance on various open-vocabulary categories, making it unsuitable for the proposed OCMOT problem. Therefore, there is an urgent need for a benchmark with rich categories and abundant videos to support OCMOT. Thus, we propose a new benchmark, OCTrackB, to effectively address the above issues.

## 3 OCTrack Benchmark

### 3.1 Problem Formulation: Open-Corpus MOT

We first provide the problem formulation of OCMOT. Given a video sequence with various objects, OCMOT aims to simultaneously achieve the localization, association and recognition tasks, thus generating a bounding box $\mathbf{b} = [x, y, w, h]$, continuous ID number $d$ (along the video) and a category $c$ for each target in the video. The annotated object categories appearing during training are defined as $\mathcal{C}^{\mathrm{b}}$, *i.e.*, the base class set. In testing, we aim to obtain the OCMOT results, *i.e.*, the object category set $\mathcal{C}^{\mathrm{open}}$ is an open corpus. Obviously we have $\mathcal{C}^{\mathrm{b}} \subset \mathcal{C}^{\mathrm{open}}$, and we define the novel class set as $\mathcal{C}^{\mathrm{n}} = \mathcal{C}^{\mathrm{open}} \backslash \mathcal{C}^{\mathrm{b}}$. Note that, we take the category recognition task as a generative task, with no need for the category list of $\mathcal{C}^{\mathrm{open}}$ as input during testing. Ideally, $\mathcal{C}^{\mathrm{open}}$ contains all the categories in the real world. In practice, for OCMOT evaluation, we can limit $\mathcal{C}^{\mathrm{open}}$ to a large-scale thesaurus.

### 3.2 Principle of Benchmark Construction

To build the OCMOT benchmark (OCTrackB), we first establish the following principles:

**P0: Principle of standardness.** Following the base and novel class division mode proposed in LVIS;

**P1: Category enrichment principle.** Base/novel classes should be diverse and balanced;

**P2: Sample enrichment principle.** Evaluation videos/objects for all classes should be abundant;

**P3: Semantic compatibility principle.** The evaluation of object recognition should be compatible.

The first principle **P0** ensures the base and novel class division in our dataset is consistent with that in the widely used LVIS. This is because that previous works, *e.g.*, many open-vocabulary detection methods [53, 54, 55, 56, 57], and the open-vocabulary tracker OVTrack all use LVIS as the training dataset. As a testing dataset, OCTrackB with the same base/novel class division is more convenient for evaluating the algorithms trained on LVIS. Both **P1** and **P2** guarantee the richness of the dataset, which aims to increase the object categories and the sample amount in the dataset. This is significant for the open-corpus tracking task. The last principle **P3** aims to address the semantic ambiguity problem, which stems from two aspects. The first aspect arises from the dataset annotation. Due to

the large number of categories, the granularity of the category annotations in the dataset is misaligned, leading to inaccurate evaluations. For example, the granularity of classification for some targets is only up to 'bird', while for others, it is more specific, such as 'goose' or 'duck'. This misalignment in labeling makes it difficult to compare recognition accuracy across algorithms during evaluation. The second aspect comes from the task OCMOT. Different from the classification head in OVMOT, the proposed OCMOT handles recognition as a generative problem. This may cause semantic synonymy or subordination. For example, 'cab' and 'taxi' usually mean the same thing, which are both types of 'car'. Therefore, for the ground truth 'cab', the prediction result of 'taxi' or even 'car' should be reconsidered and not simply taken as false. As shown in the example in Figure 2(a), we divide the categories in LVIS into multiple levels and conduct multi-level evaluations to strive to achieve Principle **P3**. Following the above principles, we build OCTrackB.

### 3.3 Dataset Collection and Annotation

By investigating the recent video datasets, we select two large ones with various object categories, *i.e.*, TAO [11] and LV-VIS [58], as the basis for constructing OCTrackB. TAO is a generic-category object tracking dataset, with a total of 833 classes and 2,907 videos. LV-VIS is a large-vocabulary video instance segmentation dataset with 1,196 classes and 4,828 videos. Previous work OVTrack [15], also following **P0**, directly uses TAO's validation and test sets (annotations from BURST [59]) and only maintains the classes that overlap with LVIS for data selection, and form the OVTrack testing datasets, *i.e.*, OVTAO validation (OVTAO-val) and test (OVTAO-burst) sets. The simple category intersection operation decreases the number of classes significantly. In this work, we consider the advantages of TAO, which provides longer videos but has a limited number of categories (overlapped with LVIS). On the other hand, the LV-VIS dataset offers a larger number of object categories, which can effectively compensate for the shortcomings of TAO, making it more suitable for addressing the OCMOT task. Specifically, we filter the test and validation sets of TAO and the training and validation sets of LV-VIS to select videos that meet principle **P0**. To satisfy **P1**, we use a greedy algorithm aiming to minimize the total number of videos while ensuring that each category contains at least two videos whenever possible, thus ensuring category diversity and balance. This results in a selection of 903 videos covering 892 categories (also contained in LVIS). To meet **P2**, we again use a greedy algorithm, aiming to allocate as many tracks as possible for each category while maintaining the same total number of videos. This further produces 732 videos with 4,766 tracks. In total, we collected 1,635 videos, of which 496 include novel category objects and 1,600 encompass base categories.

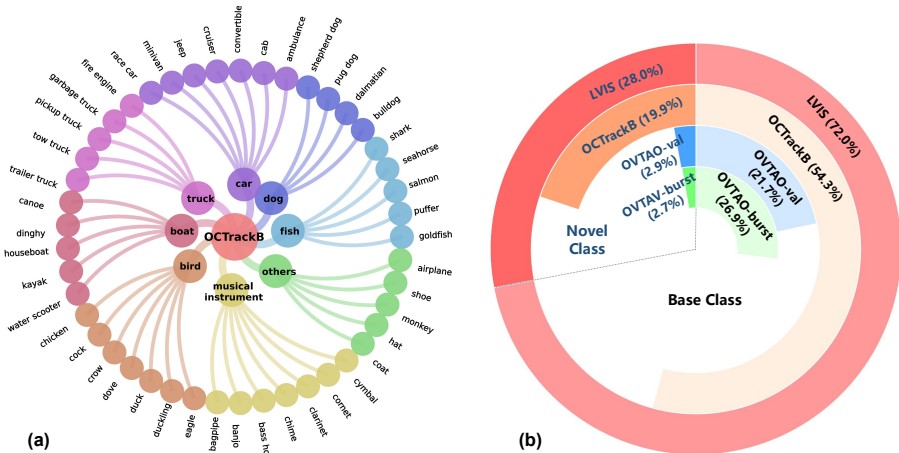

Figure 2: Statistics and comparison of the object categories appearing in the datasets.

### 3.4 Dataset Statistics and Comparison

We then show the statistics of OCTrackB and compare it with two existing OVTrack datasets, *i.e.*, OVTAO-val [15] and OVTAO-burst [59]. OCTrackB has the following typical advantages.

**Various and balanced object categories.** OCTrackB contains a total of 892 available categories, composed of 653 base and 239 novel classes. We show some example classes in OCTrackB in Figure 2(a), the categories cover every aspect in the real-world applications, *e.g.*, various transportation, animals, and household items, *etc*. Note that, following the original class annotation in our

basic datasets TAO and LV-VIS, OCTrackB involves multi-granularity categories. For example, the fine-grained class 'shepherd dog' and its general class 'dog' are concomitant in OCTrackB's category list. We leverage this subordinate relation to design the new evaluation metric in the following section.

As shown in Figure 2(b), OCTrackB includes 653 base and 239 novel classes, which account for 75.5% of the original LVIS base categories and 70.9% of the novel categories, respectively, effectively ensuring the category diversity. For previous datasets, OVTAO-val and OVTAO-burst contain 30.1% and 37.4% of the original LVIS base classes, respectively. With respect to the novel class, the ratios are only about 10% (2.9%/2.7% *vs.* 28.0%). The various object categories make OCTrackB more comprehensive in evaluating open-corpus object tracking performance.

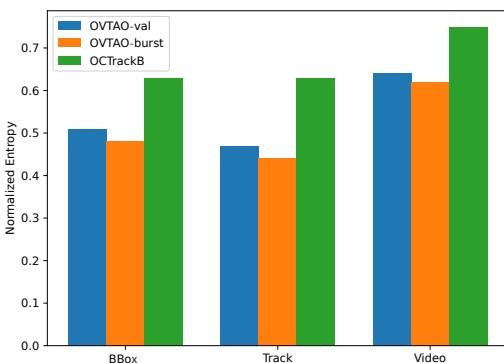

Figure 3: Normalized entropy of different units.

Next, we consider the category balance of the dataset. As shown in Figure 3, we calculate the normalized entropy of different units (object boxes, object tracks, videos) and the category set. Specifically, for $N$ categories in the dataset, we compute the Shannon Entropy as $H(p) = -\sum_{i=1}^{N} p_i \log(p_i)$, where $p_i$ denotes the probability of a unit belonging to category $i$, and the Maximum Entropy as $H_{\max} = \log(n)$. Then we get the Normalized Entropy as $\text{NE} = \frac{H(p)}{H_{\max}}$, which can reflect the category balance in the dataset. We can see that, the class balance of the proposed OCTrackB is higher than OVTAO-val and OVTAO-burst. We know that, in the real world, the object category distribution is long-tail but not balanced. However, as an evaluation benchmark, we try to keep the category balanced to guarantee that the evaluation is not dominated by the large-scale yet simple classes.

**Abundant samples for both base and novel classes.** As shown in Figure 4, we show the number of objects, tracks, and videos in OVTAO-val, OVTAO-burst, and OCTrackB datasets. The statistics are split through the base and novel classes. We can see that, for the base class, the number of object boxes, tracks, and videos in OCTrackB is greater than those of OVTAO-val and OVTAO-burst. Moreover, in terms of novel class, we can see that the data amount of OCTrackB is significantly larger than that of OVTAO-val and OVTAO-burst, with the increase ranging from 7.7 to 11.2 times.

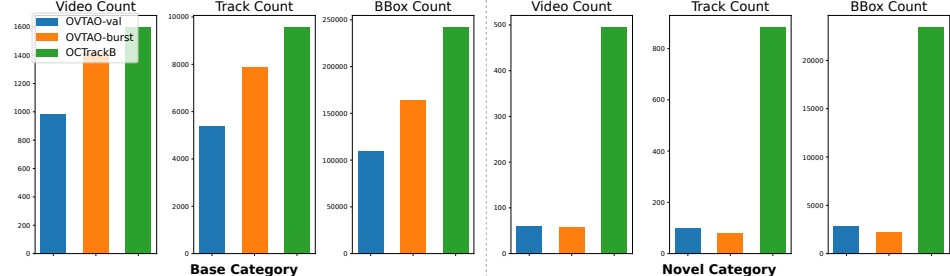

Figure 4: Statistics of the videos, track, and objects for base/novel classes in different datasets.

From the comparison, we can see that the proposed OCTrackB is more in line with the above principles **P1** and **P2**. We further provide more statistics of OCTrackB to show its data distribution and characteristics in the *supplementary material*.

## 3.5 Evaluation Metrics

Following [15], we use the open-category tracking metric namely tracking-every-thing accuracy (TETA) in [13] for evaluation. TETA is composed of three parts, *i.e.*, object localization, association, and classification accuracies. First, the localization accuracy (LocA) is computed through the matching of the GT boxes with predicted boxes without considering class, as $\text{LocA} = \frac{|\text{TPL}|}{|\text{TPL}|+|\text{FPL}|+|\text{FNL}|}$. Second, association accuracy (AssocA) is determined by matching the identities of associated GT instances with the predicted association, as $\text{AssocA} = \frac{1}{|\text{TPL}|} \sum_{b \in \text{TPL}} \frac{|\text{TPA}(b)|}{|\text{TPA}(b)|+|\text{FPA}(b)|+|\text{FNA}(b)|}$. Finally, classification accuracy (ClsA) is calculated using all correctly localized instances, by com-

paring the predicted classes with the corresponding GT classes, as $\text{ClsA} = \frac{|\text{TPC}|}{|\text{TPC}|+|\text{FPC}|+|\text{FNC}|}$. The TETA score is computed as the mean value of the above three scores as $\text{TETA} = \frac{\text{LocA}+\text{ClsA}+\text{AssocA}}{3}$.

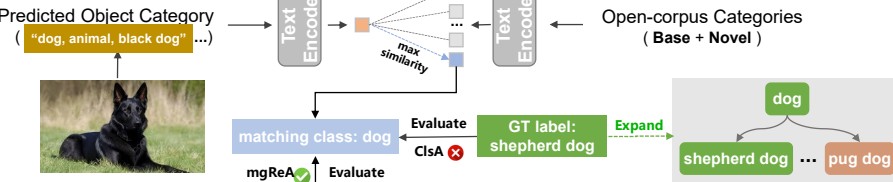

Figure 5: Illustration of the multi-granularity evaluation metric.

In previous open-category tracking tasks [45, 15], object recognition is always taken as a classification problem using the above ClsA metric. We take the recognition as a generative task, which may generate multiple labels. This way, as shown in Figure 5, we first use CLIP [16] to encode the predicted output (multiple generated object categories concatenated into a single prompt using ',') and each base/novel category in LVIS. Next, we calculate the similarity between these encoded features to choose a high-similarity category label, *i.e.*, the matching class (a single class name in LVIS), which can be used to compute ClsA. Note that, the base and novel categories are only used for result evaluation, which is different from OVMOT that uses them to generate the prediction results.

As discussed in **P3** at Section 3.2, the open-corpus tracking may introduce the semantic ambiguity problem. To address this problem, we design a multi-granularity recognition accuracy (mgReA). Specifically, considering the diversity of the generated vocabulary, we aggregate the categories in LVIS according to WordNet [60] as a hierarchy structure. As shown in Figure 5, when computing mgReA, if the ground-truth category label belongs to any category within this aggregated multi-granularity class hierarchy, it is considered an expanded successful recognition. A simple example is that, for the ground-truth label 'shepherd dog', we expand it to 'dog'. For the matching class (prediction) of 'dog', ClsA will judge it as a false result, but mgReA takes it as true. This metric provides a more intuitive and compatible evaluation, since we do not need very fine-grained classifications in many cases. Based on mgReA, we define a new comprehensive metric called tracking&recognizing-every-thing accuracy (TRETA) as $\text{TRETA} = \frac{\text{LocA}+\text{mgReA}+\text{AssocA}}{3}$ for the OCMOT problem.

## 4  A Baseline Method: OCTracker

**1) Localization:** As shown in Figure 6, similar to most tracking-by-detection based MOT approaches, we first need to obtain object bounding boxes for each frame. Since our focus is on open-corpus object tracking, we aim to localize generic-class objects. We employ the well-known detector Deformable DETR [61] as the basic network structure of the localization head. Deformable DETR uses Hungarian matching to map the predicted detections to the ground truths, and then aligns the corresponding boxes through category classification loss and bounding box regression loss. In our framework, we do not consider the object class in the localization head, aiming to train a class-agnostic object detector. This way, we replace the category classification loss in [61] with a binary cross-entropy loss, *i.e.*, to estimate whether a region candidate is an object of interest or not.

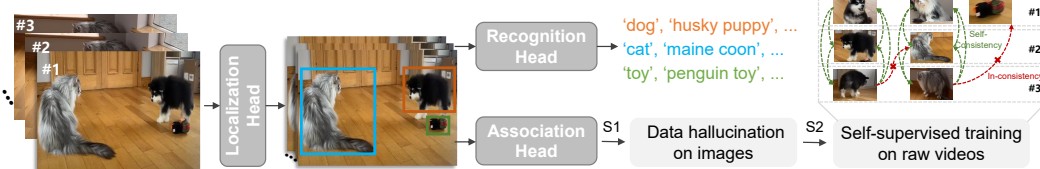

Figure 6: Pipeline of the proposed baseline method OCTracker.

**2) Recognition:** The recognition head is used to generate the category name of the object. It mainly consists of a generative language model, for which we use FlanT5-base [62] and initialize it with its pre-trained weights. The visual features of the candidate objects obtained from Deformable DETR are mapped to the input space of the generative model through a projection layer, and then processed by a generative encoder and decoder, both composed of self-attention layers and feedforward neural networks. The encoder's output interacts with the decoder through the cross-attention layers. Then the decoder's output is passed through a softmax layer to predict the corresponding word, while the

prediction of the previous word is used as input for training the next word prediction. The generative model is trained following the manner and loss function in [63], using the VG [64] and GRIT [65] image-text pairs as training data. The beam size of the language model is controllable. We set it to 2, meaning that we generate two category nouns for each object.

**3) Association:** As a tracking task, a key step is to associate objects along the video. For this purpose, we consider a two-stage training strategy to train the object similarity learning model for association. Since there is no large-scale generic-object video dataset with tracking annotations [15], we can only use the image datasets or raw videos for training. The first stage is to learn the association model with static images. Following [15], we apply the data hallucination strategy to generate the pairwise images for training. Specifically, given an image of base categories in LVIS [17], we use a diffusion model to generate its adjoint image with the same object categories but different styles. Then the similarity learning can be achieved by contrastive learning between each image pair, in which the same object as the positive sample, other objects, and generated objects as negative samples. The second stage is to learn the association model with raw videos. Following [66], we employ a self-supervised strategy to learn object similarity using the raw videos in TAO training set. Specifically, as shown in Figure 6, given a reference object in a frame, we first seek its most similar target in another frame. Then from this target, the most similar object in the original frame should be the reference object. Based on this object self-similarity rationale, we use the self-supervised losses [66] to learn the object similarity. More details and limitations of OCTracker are discussed in the *supplementary material*.

# 5 Experimental Results

## 5.1 Comparison Methods

As a new problem, there is no approach that can directly handle the OCMOT. We try to include as many approaches as possible with necessary modifications for comparison on the proposed OCTrackB. ❶ First, we select two strong MOT algorithms, *i.e.*, QDTrack [20] and TETer [67]. The classical MOT approaches can not handle the object recognition task in OCTrack, thus we train both algorithms on both base and novel categories in LVIS [17] and TAO [11] training sets by a closed-set training approach and then evaluate their performance on the OCTrackB. ❷ Second, we include the only public open-vocabulary MOT algorithm OVTrack [15] in the experiments, by additionally giving the base and novel class list during testing as the setting of it. ❸ Also, to evaluate more related approaches, we further use the way of approach combination. Specifically, we select an open-vocabulary detection (OVD) algorithm for object localization and recognition (classification) combined with an object tracking method for association, to achieve the OCMOT. We select three state-of-the-art OVD algorithms, *i.e.*, VLDet [55], CoDet [54], MM-OVOD [53], and three tracking methods including the appearance-based tracking method namely DiffuTrack in OVTrack [15], motion-based tracking in ByteTrack [1] and OC-SORT [2]. Note that, these methods also need the base and novel class list for testing. ❹ Finally, we employ an open-ended generative object detection method namely GenerateU [63] as the detector, combined with the above tracking modules, for the OCTrack task. This series of methods is really under the setting of OCTrack without the class lists when testing. ❺ The proposed baseline method OCTracker is also included for comparison.

## 5.2 Benchmark Results

**Comparison among state-of-the-art methods.** As shown in Table 1, we can see that, the classical MOT algorithms QDTrack [20] and TETer [67] provide a satisfied performance on the localization and association tasks since these methods are specifically designed for such tasks and they have been trained on both base-class and novel-class data. However, we can see that the object recognition results, *i.e.*, the ClsA score, are very poor. This is because these methods can not handle the diverse long-tailed classification with up to 892 categories in OCTrackB. Also, ClsA metric only considers top-1 classification accuracy, but these classes are fine-grained. From this point, the proposed recognition score mgReA is more reasonable. Then we can see that the open-vocabulary tracking approach OVTrack [15] provides relatively good results among all competitors. However, it uses the class list as input during training, the open type is set as OV. Under the same setting of OV, we select three more recent OV detection methods VLDet [55], CoDet [54] and MM-OVOD [53] with three classical tracking strategies for association. Among them, DiffuTrack uses a diffusion model based data hallucination strategy [15] to learn the object similarity for association. ByteTrack [1] applies a

detection selection strategy and uses the motion feature for the association. OC-SORT [2] further considers the occlusion when using the motion feature. For the above combination-based methods, we find that their overall performance is comparable with OVTrack. In terms of the comprehensive TETA score, CoDet [54] and MM-OVOD [53] with DiffuTrack outperform OVTrack on the base class. VLDet [55] and CoDet [54] with DiffuTrack outperform OVTrack on the novel class. But the margins are all not very large. Note that, the proposed OCTrackB *can also be used for the open-vocabulary MOT problem* as shown in the above results. However, in this work, *we are more interested in the proposed OCMOT problem*, which is more practical and promising.

Next, we present the results under the OC setting, in which we use the alone detector following the OC setting, *i.e.*, GenerateU [63] with the above three tracking strategies to implement the OCMOT. We report the results of them, and also the proposed OCTracker, at the bottom of Table 1. We can see that, in terms of the object recognition task using the ClsA metric, OCTracker provides a comparable result with other approaches since the underlying language models used for object recognition (class generation) are similar. OCTracker also provides better association results (AssocA) for both base and novel classes, which demonstrates the advantages of the association head in OCTracker.

Table 1: Comparison results on the proposed OCTrackB (%).

| Methods | Train Data | | Open Type | Base Class | | | | | | Novel Class | | | | | |
|---|---|---|---|---|---|---|---|---|---|---|---|---|---|---|---|
| | Base | Novel | OV/OC | TETA | LocA | AssocA | ClsA | mgReA | TRETA | TETA | LocA | AssocA | ClsA | mgReA | TRETA |
| QDTrack [20] | ✓ | ✓ | - | 26.6 | 32.2 | 38.8 | 8.8 | 13.7 | 28.2 | 28.1 | 37.8 | 46.4 | 0.1 | 7.6 | 30.6 |
| TETer [67] | ✓ | ✓ | - | 26.5 | 36.7 | 41.5 | 1.2 | 3.4 | 27.2 | 31.4 | 45.4 | 48.7 | 0.1 | 2.5 | 32.2 |
| OVTrack [15] | ✓ | † | OV | 34.6 | 37.8 | 44.3 | 21.7 | 28.9 | 37.0 | 32.8 | 44.1 | 50.6 | 3.6 | 12.3 | 35.7 |
| VLDet [55] | | | | | | | | | | | | | | | |
| + DiffuTrack [15] | ✓ | † | OV | 32.9 | 36.1 | 45.2 | 17.5 | 25.4 | 35.6 | 32.9 | 40.9 | 49.5 | 8.2 | 15.1 | 35.2 |
| + ByteTrack [1] | ✓ | † | OV | 29.3 | 32.0 | 41.6 | 14.4 | 19.8 | 31.1 | 29.5 | 34.5 | 48.0 | 6.0 | 12.0 | 31.5 |
| + OC-SORT [2] | ✓ | † | OV | 26.1 | 29.2 | 36.1 | 13.1 | 18.1 | 27.8 | 27.0 | 34.1 | 40.5 | 6.5 | 12.5 | 29.0 |
| CoDet [54] | | | | | | | | | | | | | | | |
| + DiffuTrack [15] | ✓ | † | OV | 35.1 | 36.7 | 46.3 | 22.4 | 29.3 | 37.4 | 33.0 | 39.2 | 48.0 | 11.8 | 18.4 | 35.2 |
| + ByteTrack [1] | ✓ | † | OV | 31.4 | 33.3 | 42.8 | 18.1 | 23.9 | 33.3 | 31.5 | 37.2 | 47.3 | 10.2 | 16.5 | 33.7 |
| + OC-SORT [2] | ✓ | † | OV | 28.7 | 31.0 | 37.5 | 17.5 | 22.9 | 30.5 | 28.5 | 35.3 | 40.2 | 9.9 | 15.8 | 30.4 |
| MM-OVOD [53] | | | | | | | | | | | | | | | |
| + DiffuTrack [15] | ✓ | † | OV | 35.4 | 38.6 | 47.5 | 20.0 | 26.3 | 37.5 | 32.6 | 42.5 | 51.1 | 4.3 | 6.8 | 33.5 |
| + ByteTrack [1] | ✓ | † | OV | 31.4 | 33.0 | 43.2 | 18.0 | 23.8 | 33.3 | 29.7 | 36.8 | 47.7 | 4.6 | 11.0 | 31.8 |
| + OC-SORT [2] | ✓ | † | OV | 27.3 | 29.3 | 36.8 | 15.8 | 20.9 | 29.0 | 25.0 | 32.0 | 38.6 | 4.5 | 10.2 | 26.9 |
| GenerateU [63] | | | | | | | | | | | | | | | |
| + DiffuTrack [15] | ✓ | ✗ | OC | 31.0 | 37.5 | 40.3 | 15.2 | 21.0 | 32.9 | 29.5 | 43.4 | 43.5 | 1.6 | 9.3 | 32.1 |
| + ByteTrack [1] | ✓ | ✗ | OC | 28.2 | 30.7 | 38.7 | 15.2 | 20.7 | 30.0 | 27.1 | 36.8 | 42.8 | 1.7 | 9.8 | 29.8 |
| + OC-SORT [2] | ✓ | ✗ | OC | 27.0 | 28.8 | 37.4 | 14.8 | 20.2 | 28.8 | 25.1 | 32.7 | 40.7 | 1.8 | 10.1 | 27.8 |
| OCTracker | ✓ | ✗ | OC | 32.2 | 38.8 | 42.2 | 15.6 | 21.1 | 34.0 | 31.5 | 45.7 | 46.2 | 2.5 | 10.5 | 34.1 |

'✓' denotes using the corresponding videos of base/novel class, '†' denotes only using the class list but not the videos for testing, and '✗' means using nothing about the novel class.

**Results of new metrics.** We then discuss the results of using different recognition metrics, *i.e.*, the previous ClsA and proposed mgReA. We can see first that, in most cases, mgReA provides the consistent evaluation as ClsA, *i.e.*, better ClsA leads to better mgReA. This verifies the availability of mgReA that can correctly reflect the object recognition performance. Also, we can see that the margin between two mgReA scores is generally larger than that of two ClsA scores. This means mgReA can better reflect the gap among different approaches. Especially when computing the TETA score, if the recognition score (using ClsA) is similar, the TETA will be dominated by the other two metrics (LocA and AssocA). This way, the proposed TRETA uses a more discriminative mgReA score for better evaluation. A special case is shown in the first two lines, for novel class set, QDTrack [20] and TETer [67] provide the same result when using ClsA (0.1 *vs.* 0.1) without discriminability. But the mgReA metric (7.6 *vs.* 2.5) can effectively evaluate their performance.

### 5.3 In-depth Analysis

**Discussion and insights.** From Table 1, we can observe that the performances of all the methods are generally poor, especially for the recognition task. This reflects *the challenges of OCTrackB and also the OCMOT problem*, which have great space for improvement. We further compare the results generated by different settings of OVMOT and OCMOT, by taking the OVTrack and OCTracker for example. We also find that the object recognition performance of OCTracker is invariably lower than OVTrack using either ClsA or mgReA on both base and novel classes. This is reasonable since OCTracker no longer requires the (base and novel) class list used in OVTrack. Although the OV-based methods, overall speaking, perform better than OC-based methods, *the performance gap between them is not large*. This demonstrates that *the proposed more practical OCMOT task is very promising*.

**Dataset comprehensiveness.** As discussed above, both OVTAO and the proposed OCTrackB use the videos in the TAO dataset. We select the overlapped videos in OVTAO and OCTrackB, and apply the

public OVTrack [15] method on them for comparison. As shown in Table 2, we find that although the overlapped videos included in OCTrackB represent approximately 41% of the original OVTAO-val or OVTAO-burst, the experimental results show negligible differences. The evaluated results on both base and novel categories diverge by no more than 0.6% for TETA, TRETA, when compared to the original OVTAO dataset. This comparison demonstrates that the portion of OVTAO dataset included in OCTrackB *is highly representative, containing the data distribution diversity of the original OVTAO dataset*. Besides these videos, OCTrackB also includes the data from LV-VIS. The richness of categories and quantity of samples has been significantly expanded in terms of principles **P1** and **P2**, making OCTrackB highly effective and comprehensive for the OCMOT.

Table 2: Comparison results on datasets extracted from OVTAO in OCTrackB (%).

| Dataset | # Video | Base Class | | | | | | Novel Class | | | | | |
|---|---|---|---|---|---|---|---|---|---|---|---|---|---|
| | | TETA | LocA | AssocA | ClsA | mgReA | TRETA | TETA | LocA | AssocA | ClsA | mgReA | TRETA |
| OVTAO-val | 993 (100%) | 35.5 | 49.3 | 36.9 | 20.2 | 29.2 | 38.5 | 28.0 | 48.8 | 33.6 | 1.5 | 9.7 | 30.7 |
| OCTrackB$_{OVTAO-val}$ | 402 (40.5%) | 36.1 (Δ0.6) | 50.2 | 37.8 | 20.4 | 29.3 | 39.1 (Δ0.6) | 27.8 (Δ0.2) | 46.4 | 35.7 | 1.2 | 8.3 | 30.1 (Δ0.6) |
| OVTAO-burst | 1428 (100%) | 32.0 | 45.6 | 33.5 | 16.9 | 24.1 | 34.4 | 24.4 | 42.3 | 29.1 | 1.8 | 6.1 | 25.8 |
| OCTrackB$_{OVTAO-burst}$ | 585 (41.0%) | 32.1 (Δ0.1) | 45.5 | 34.4 | 16.4 | 24.0 | 34.6 (Δ0.2) | 25.0 (Δ0.6) | 43.1 | 29.7 | 2.3 | 6.4 | 26.4 (Δ0.6) |

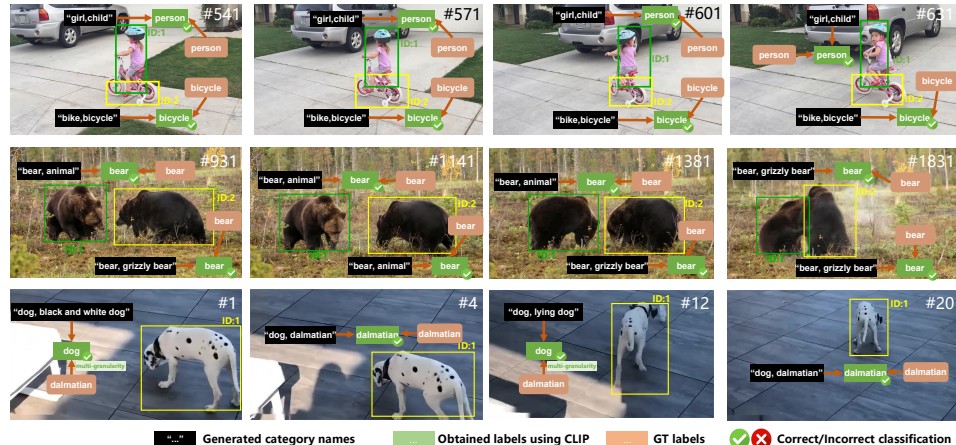

Figure 7: Illustration of the qualitative results of OCTracker.

**Visualization analysis.** Figure 7 presents some visualization results of OCTracker, in which bounding boxes with the same color indicate the same track ID, text boxes with a black background display the generated category names (prediction), while the text boxes with a green background show the labels obtained using CLIP for evaluation, and the text boxes with a brown background indicate the ground-truth labels in the dataset. We can see that OCTracker encapsulates a rich understanding of object categories. For instance, in the first row, OCTracker is not only able to identify the object as a 'child', but also recognizes it as a 'girl', thereby providing a more comprehensive description of the target. Importantly, this is achieved without the need for any pre-specified category restrictions. In the second row of results, the generated output includes the prediction 'grizzly bear', even more specific than the ground truth 'bear'. The third row demonstrates the effectiveness of the proposed mgReA in Section 3.5. We can observe that for tracking a specific subclass 'dalmatian' of 'dog', OCTracker effectively describes the target's characteristics, such as 'black and white dog'. It can also predict its super-category 'dog' and accurately identify the subclass 'dalmatian' in certain frames. When the target is recognized as 'dog', the multi-granularity metric mgReA traces back to the expanded label 'dog' from the ground-truth label 'dalmatian', effectively addressing the misalignment between the generated results and GT labels. More visualizations can be found in the *supplementary material*.

## 6 Conclusion

In this work, we have proposed a novel yet practical problem of OCMOT. We build a large-scale and comprehensive benchmark OCTrackB, to provide the standard evaluation platform for this problem. Compared to similar competitor datasets, OCTrackB has the advantage of containing more diverse and balanced object categories, and significantly more testing samples for both base and novel classes, especially the novel. Besides the dataset, we also design a new multi-granularity recognition metric to alleviate the semantic ambiguity problem for object recognition. Extensive benchmark evaluations for numerous state-of-the-art methods have demonstrated the rationale of the proposed OCMOT problem, and the usefulness of the OCTrackB benchmark.

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
