# OCTrack: Benchmarking the Open-Corpus Multi-Object Tracking
## *Supplementary Material*

## A    Access to Dataset and Benchmark

The dataset, related benchmark and Croissant metadata in this paper can be found via the link: OCTrack Benchmark, which will be released to the public when the paper is accepted.

## B    Limitation Discussion

We discuss the limitations of the proposed OVMOT problem and OCTracker.

First, the OCMOT task addresses the limitations of closed-set MOT tasks with limited object categories and the constraint of OVMOT requiring specific categories. However, the accuracy of the results in this task generally lags behind those of the classical MOT tasks. Therefore, this practical task needs more following algorithms to improve it.

Second, we found that although the association mechanism can effectively track targets in many cases, it generates different category nouns for the same track ID, resulting in a lack of consistency and coherence in the generated object categories. As shown in Figure 1(a), the generated categories for this dog include 'black and white dog,' 'dalmatian,' and 'lying dog' in the same track, which are not consistent. Such changes in the predicted object categories during tracking make it difficult for the tracker to accurately maintain a stable category. To address this problem, in future work, we plan to study the mutual collaboration mechanism between tracking and recognition in OCMOT, *i.e.*, using continuous tracking to help the consistent category prediction, as well as making use of the category estimation results to assist object tracking.

Third, as discussed in the 'evaluation metrics' in the main paper, due to the diversity of generated target category nouns, we use the CLIP to match these nouns with the most similar label from the GT category set. The rationale behind this approach is to give a more appropriate evaluation, through comprehensive semantics matching with the help of pre-trained multi-modal models. This metric is reasonable. However, it may not be the optimal solution. We find that sometimes the correct category noun is present among the generated nouns, but the CLIP-based matching still classifies it into a more general category. As shown in Figure 1 (b), we can see that although the generated target category by our method includes 'pug dog,' the final label matched by CLIP used for evaluation is 'dog.' Since the ground-truth label is 'pug dog', if we directly use this matching label 'dog' for judging, the evaluation result will be 'false'. But, fortunately in this case, if we use the multi-granularity recognition (mgReg) metric, the evaluation result will become 'true'. Even though, we still think that the overall evaluation system for OCMOT has space for improvement.

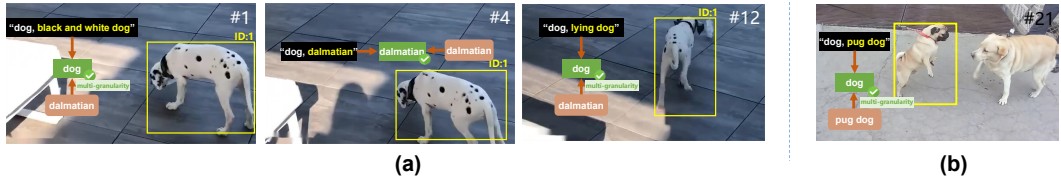

(a)                                                                                          (b)

Figure 1: Qualitative analysis of limitations.

We have discussed some issues that were not addressed in this work. As the first OCMOT benchmark, we aim to promote the expansion of tracking tasks with various constraints in existing works to more general and practical tracking. This will significantly advance the progress of the tracking community. We hope that with our efforts or those of other researchers in the future, these limitations can be partly alleviated or effectively resolved.

## C    Failure Case Study

We also present a representative failure case in Figure 2. This failure example includes an ID switch caused by target appearance differences, and the misclassification of similar categories. Specifically,

the target with ID 1 at frame # 541 is failed to be tracked at # 631. This may be caused by the irregular motion and shape variation during this period. For recognition, the 'deer' appearing in this video is incorrectly recognized as the 'sheep'. This is because they look very similar, and the viewing angle makes the video not contain enough discriminative characteristics. From this point, while acknowledging the practical value of OCTracker, we also recognize the ample room for improvement in addressing the various challenges in the OCMOT problem.

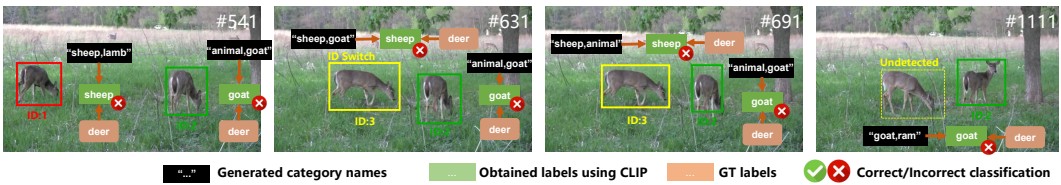

Figure 2: Qualitative analysis of a failure case.

# D    Dataset Comparison

We further provide more statistics of OCTrackB to show its data distribution and characteristics in Table 1. We can see that regardless of the total number of videos, frames, tracks, and bounding boxes, OCTrackB is larger than the other two datasets. The 'Density' metric means the object count in each frame, where OCTrackB exceeds the other two datasets significantly. Additionally, for the average number of tracks per video, and the average number of boxes per video or track, OCTrackB is also superior. These comparisons fully reflect the enrichment of OCTrackB, which is beneficial for the evaluation of the open-corpus-related task and tracking tasks.

Table 1: Comparison of different datasets on various factors.

| Datasets | # Vid. | # Frm. | # Track | # BBox | Density | Track/Vid. | BBox/Vid. | BBox/Track |
|----------|--------|--------|---------|--------|---------|------------|-----------|------------|
| OVTAO-val | 998 | 36K | 5K | 113K | 1 ~11 | Avg. 5.5 | Avg. 114.2 | Avg. 20.6 |
| OVTAO-burst | 1,419 | 52K | 8K | 167K | 1 ~11 | Avg. 5.6 | Avg. 117.5 | Avg. 21.0 |
| OCTrackB | 1,635 | 55K | 10K | 266K | 1 ~41 | Avg. 6.4 | Avg. 162.7 | Avg. 25.4 |

# E    Implementation Details of OCTracker

In this section, we provide more implementation details of the OCTracker. In the localization module, we use Swin Transformer [1] as the backbone for visual encoding. Following [2], the Deformable DETR architecture has 6 encoder layers and 6 decoder layers. The number of object queries is set to 300.

In the recognition module, we use the FlanT5-base [3] and initialize it with its pre-trained weights. Flan T5-Base is a Transformer model that includes both encoder and decoder structures. We select this network due to its parameter count of only 250M, which effectively reduces computational complexity. Additionally, its performance is significantly enhanced through efficient multi-task instruction fine-tuning. The beam size is set to 2, *i.e.*, the output number of categories of each object is 2.

In the association module, the data hallucination strategy uses the same setting as OVTrack [4] with a learning rate of $2 \times 10^{-3}$. In the self-supervised strategy, we use the TAO training dataset [5] without any annotation for training. We select a continuous sequence of 24 frames as a batch, grouping them using $C_N^2$ and $C_N^3$ combinations to enhance data scale. Then, we employ the cyclic-consistency loss proposed in the work [6] for self-supervised training with a learning rate of $2 \times 10^{-4}$.

In the training stage, we first train the localization and recognition modules together for 20 epochs. The learning rate of the backbone and detection head is set to $2 \times 10^{-4}$, and the language model is set to $3 \times 10^{-4}$. Then we train the association head using a hallucination strategy for 6 epochs and self-supervise the association head for 14 epochs. In the inference stage, we associate the historical tracks to the objects detected in the current frame using the appearance feature similarity obtained from the trained association head. Following the methodology in [4], we compute the

similarity between historical tracks and detected objects using both bi-directional softmax [7] and cosine similarity metrics. In line with traditional MOT methods, we assign an object to a track if the similarity score surpasses a matching threshold. If an object doesn't match any existing track, a new track is initiated if its detection confidence score from the classification head exceeds a threshold; otherwise, it is ignored.

We use PyTorch as the implementation framework and conduct the experiments on a server with 8 RTX 3090 GPUs. The optimizer used in our method is AdamW [8] with a weight decay of 0.05.

## F  Annotation Format

To make the dataset easier to use, we organize it in the open and widely used COCO format, allowing it to be used directly with frameworks like Detectron2 and MMDetection without any modification.

```
{
  "info": {...},
  "categories": [
    {
      "image_count": 4,
      "synonyms": [
        "aerosol_can",
        "spray_can"
      ],
      "def": "a dispenser that holds a substance under pressure
        ",
      "id": 1,
      "synset": "aerosol.n.02",
      "name": "aerosol_can",
      "frequency": "c",
      "instance_count": 4
    },
    ...
  ],
  "tracks": [
    {
      "id": 11747,
      "category_id": 557,
      "video_id": 1848,
    },
    ...
  ],
  "videos": [
    {
      "id": 1848,
      "width": 1280,
      "height": 720,
      "name": "tao/frames/val/LaSOT/bird-7/",
      "neg_category_ids": [1130,408,869,786,409]
    },
    ...
  ],
  "images": [
    {
      "id": 70064,
      "video": "tao/frames/val/LaSOT/bird-7",
      "_scale_task_id": "5dad734fa6e4543773ab0f8d",
      "width": 1280,
      "height": 720,
      "file_name": "tao/frames/val/LaSOT/bird-7/00000871.jpg",
```

```
        "frame_index": 870,
        "video_id": 1848,
        "frame_id": 0,
        "neg_category_ids": [1130, 408, 869, 786, 409],
    },
    ...
  ],
  "annotations": [
    {
      "segmentation": [
        [416, 84, 1214, 84, 1214, 639, 416, 639]
      ],
      "bbox": [416.0, 84.0, 798.0, 555.0],
      "area": 442890,
      "iscrowd": 0,
      "id": 259058,
      "image_id": 70064,
      "category_id": 557,
      "track_id": 11747,
      "scale_category": "pet",
      "video_id": 1848,
      "instance_id": 11747
    },
    ...
  ]
}
```

## G  Datasheet for Dataset

We follow [9] for writing the data sheet of the proposed OCTrackB.

1. Motivation

   (a) **For what purpose was the dataset created?**
       First, the first proposed OCT task requires a diverse dataset with a rich variety of cate-
       gories, especially in the novel categories. However, existing datasets like OVTAO have
       an extremely limited number of categories, and the sample numbers for each category
       are imbalanced. To address this issue, we aim to construct a new, more comprehensive
       dataset based on the existing datasets. Second, considering the characteristics of the
       OCT task and the generative classification uncertainty, we need a more effective hi-
       erarchical classification evaluation method. Therefore, our proposed new dataset will
       effectively organize the hierarchical relationships of the categories. Third, we aim for
       the proposed dataset to be effective and experimentally validated.

   (b) **Who created this dataset?**
       The authors listed in the paper carefully and thoughtfully designed the dataset.

   (c) **Who funded the creation of the dataset?**
       This work was supported in part by the NSFC under Grants 62072334, U1803264.

   (d) **Any other comments?**
       [NA]

2. Composition

   (a) **What do the instances that comprise the dataset represent (e.g., documents, photos,
       people, countries)? (Are there multiple types of instances (e.g., movies, users, and
       ratings; people and interactions between them; nodes and edges)? Please provide
       a description.)**
       The dataset consists of 1,635 videos and 55K frames, with annotated target categories
       including 892 classes. These annotations include their bounding boxes, track IDs, and
       classes.

(b) **How many instances are there in total (of each type, if appropriate)?**
There are 266K instances in total. Please see Table 1 for details.

(c) **Does the dataset contain all possible instances or is it a sample (not necessarily random) of instances from a larger set?**
The dataset contains as many instances as possible in each video.

(d) **What data does each instance consist of? ("Raw" data (e.g., unprocessed text or images) or features? In either case, please provide a description.)**
The dataset includes 2D bounding boxes, labels of classification, and track IDs for each target in the videos. More details can be seen in Section F.

(e) **Is there a label or target associated with each instance? If so, please provide a description.**
[Yes] The labels can be seen in Section F.

(f) **Is any information missing from individual instances? (If so, please provide a description, explaining why this information is missing (e.g., because it was unavailable). This does not include intentionally removed information, but might include, e.g., redacted text.)**
[NA]

(g) **Are relationships between individual instances made explicit (e.g., users' movie ratings, social network links)? (If so, please describe how these relationships are made explicit.)**
[No]

(h) **Are there recommended data splits (e.g., training, development/validation, testing)? (If so, please provide a description of these splits, explaining the rationale behind them.)**
[No] Our dataset is only used as a test set and does not include a training set.

(i) **Are there any errors, sources of noise, or redundancies in the dataset? (If so, please provide a description.)**
Since the original dataset was manually annotated, there may be some errors.

(j) **Is the dataset self-contained, or does it link to or otherwise rely on external resources (e.g., websites, tweets, other datasets)?**
Our new dataset OCTrackB is reconstructed and organized from the TAO [5] and LV-VIS [10] datasets, which can be found in TAO dataset and LV-VIS dataset.

(k) **Does the dataset contain data that might be considered confidential?**
[No]

(l) **Does the dataset contain data that, if viewed directly, might be offensive, insulting, threatening, or might otherwise cause anxiety? (If so, please describe why.)**
[No]

(m) **Does the dataset relate to people? (If not, you may skip the remaining questions in this section.)**
[Yes]

(n) **Does the dataset identify any subpopulations (e.g., by age, gender)?**
[NA]

(o) **Is it possible to identify individuals (i.e., one or more natural persons), either directly or indirectly (i.e., in combination with other data) from the dataset? (If so, please describe how.)**
[No]

(p) **Does the dataset contain data that might be considered sensitive in any way**
[No]

(q) Any other comments?
[NA]

3. Collection Process

(a) **How was the data associated with each instance acquired?**
The bounding box, class label and track ID for each instance was acquired by human annotators.

(b) **What mechanisms or procedures were used to collect the data (e.g., hardware apparatus or sensor, manual human curation, software program, software API)? (How were these mechanisms or procedures validated?)**
The annotators use the annotation software to collect annotations. The software shows a video frame by frame, and the annotator is able to use the mouse to draw annotations.

(c) **If the dataset is a sample from a larger set, what was the sampling strategy (e.g., deterministic, probabilistic with specific sampling probabilities)?**
[Yes] Our dataset is derived from the larger TAO and LV-VIS datasets, with greedy sampling strategies detailed in Section 3.3 of the main paper.

(d) **Who was involved in the data collection process (e.g., students, crowdworkers, contractors) and how were they compensated (e.g., how much were crowdworkers paid)?**
The authors collect the data from public datasets. Their compensation comes from research funding.

(e) **Over what timeframe was the data collected? (Does this timeframe match the creation timeframe of the data associated with the instances (e.g., recent crawl of old news articles)? If not, please describe the timeframe in which the data associated with the instances was created.)**
From Jan. 2024 to May 2024.

(f) **Were any ethical review processes conducted (e.g., by an institutional review board)? (If so, please provide a description of these review processes, including the outcomes, as well as a link or other access point to any supporting documentation.)**
[NA]

(g) **Does the dataset relate to people? (If not, you may skip the remaining questions in this section.)**
[Yes]

(h) **Did you collect the data from the individuals in question directly, or obtain it via third parties or other sources (e.g., websites)?**
[NA]

(i) Any other comments? [NA]

4. Preprocessing/cleaning/labeling

(a) **Was any preprocessing/cleaning/labeling of the data done (e.g., discretization or bucketing, tokenization, part-of-speech tagging, SIFT feature extraction, removal of instances, processing of missing values)? (If so, please provide a description. If not, you may skip the remainder of the questions in this section.)**
[No]

(b) **Was the "raw" data saved in addition to the preprocessed/cleaned/labeled data (e.g., to support unanticipated future uses)? (If so, please provide a link or other access point to the "raw" data.)**
[No]

(c) **Is the software used to preprocess/clean/label the instances available? (If so, please provide a link or other access point.)**
[No]

(d) Any other comments?
[NA]

5. Uses

(a) **Has the dataset been used for any tasks already? (If so, please provide a description.)**
[No]

(b) **Is there a repository that links to any or all papers or systems that use the dataset? (If so, please provide a link or other access point.)**
[No] This is the first work to do the OCT by this dataset.

(c) **What (other) tasks could the dataset be used for?**
The dataset can be used in closed-set MOT, open-vocabulary MOT.

(d) **Is there anything about the composition of the dataset or the way it was collected and preprocessed/cleaned/labeled that might impact future uses?**
[No]

(e) **Are there tasks for which the dataset should not be used? (If so, please provide a description.)**
[No]

(f) Any other comments?
[NA]

6. Distribution

(a) **Will the dataset be distributed to third parties outside of the entity (e.g., company, institution, organization) on behalf of which the dataset was created? (If so, please provide a description.)**
[No]

(b) **How will the dataset will be distributed (e.g., tarball on website, API, GitHub)? (Does the dataset have a digital object identifier (DOI)?)**
The dataset will be made publicly available on the GitHub repository.

(c) **When will the dataset be distributed?**
The dataset will be distributed at the same time as the paper is published.

(d) **Will the dataset be distributed under a copyright or other intellectual property (IP) license, and/or under applicable terms of use (ToU)? (If so, please describe this license and/or ToU, and provide a link or other access point to, or otherwise reproduce, any relevant licensing terms or ToU, as well as any fees associated with these restrictions.)**
We only sampled and redesigned the original TAO (MIT license) and LV-VIS (GPL-3.0 license) datasets. Therefore, our license should comply with these two licenses, with the GPL-3.0 being more restrictive. As a result, the new dataset from this work, OCTrackB, should follow the GPL-3.0 license.

(e) **Have any third parties imposed IP-based or other restrictions on the data associated with the instances? (If so, please describe these restrictions, and provide a link or other access point to, or otherwise reproduce, any relevant licensing terms, as well as any fees associated with these restrictions.)**
[NA]

(f) **Do any export controls or other regulatory restrictions apply to the dataset or to individual instances? (If so, please describe these restrictions, and provide a link or other access point to, or otherwise reproduce, any supporting documentation.)**
[NA]

(g) **Any other comments?**
[NA]

7. Maintenance

(a) **Who is supporting/hosting/maintaining the dataset?**
[NA] We are using publicly available datasets, each supported by their respective teams.

(b) **How can the owner/curator/manager of the dataset be contacted (e.g., email address)?**
Authors can be contacted via email.

(c) **Is there an erratum? (If so, please provide a link or other access point.)**
[No] The later erratum will be provided on the paper GitHub page.

(d) **Will the dataset be updated (e.g., to correct labeling errors, add new instances, delete instances)? (If so, please describe how often, by whom, and how updates will be communicated to users (e.g., mailing list, GitHub)?)**
[Yes] If the TAO and LV-VIS datasets are updated, then we will synchronize our updates accordingly.

(e) **If the dataset relates to people, are there applicable limits on the retention of the data associated with the instances (e.g., were individuals in question told that their data would be retained for a fixed period of time and then deleted)? (If so, please describe these limits and explain how they will be enforced.)**
[NA]

(f) **Will older versions of the dataset continue to be supported/hosted/maintained? (If so, please describe how. If not, please describe how its obsolescence will be communicated to users.)**
[NA]

(g) **If others want to extend/augment/build on/contribute to the dataset, is there a mechanism for them to do so? (If so, please provide a description. Will these contributions be validated/verified? If so, please describe how. If not, why not? Is there a process for communicating/distributing these contributions to other users? If so, please provide a description.)**
[Yes] Errors/features can be submitted as issues/pull requests on GitHub.

(h) **Any other comments?**
[NA]