# OpenReview forum: "OCTrack: Benchmarking the Open-Corpus Multi-Object Tracking"
_NeurIPS.cc/2024/Datasets_and_Benchmarks_Track — Submitted to NeurIPS 2024 Track Datasets and Benchmarks_

### Official Review · Reviewer_oy8h · 2024-07-25
**Meaningful benchmark for the future MOT models**

**Rating:** 7
**Confidence:** 4
**Correctness:** Yes.
**Clarity:** Yes.

**Review:**

Pros:

1. This benchmark improves previous open-vocabulary MOT by requiring the model to suggest the categories, which is a natural next step.
2. The construction of the benchmark is principled and solid.
3. The baseline experiments are comprehensive and convincing.

Cons:

1. It would be nice to quantitatively show the reliability of the mgReA metric. For example, randomly select a number of results, run the CLIP matching approach and compare it with human annotators matching.
2. A minor issue: the template used by this paper is not the correct NeurIPS Dataset track review template.

Overall I believe this benchmark is meaningful for future work on MOT and recommend acceptance.

**Strengths:**

See the review section above.

**Additional Feedback:**

See the review section above.

**Documentation:**

Yes.

**Ethics:**

No issue.

**Limitations:**

Yes.

**Opportunities For Improvement:**

See the review section above.

**Relation To Prior Work:**

Yes.

**Summary And Contributions:**

The paper presents open-corpus multi-object tracking (OCMOT), an extension of OVMOT that involves identifying, tracking, and recognizing a broad range of known and unknown object categories without a predefined category list. The authors develop OCTrackB, a comprehensive benchmark for OCMOT, offering a balanced mix of object classes to reduce biases. They also introduce a new multi-granularity recognition metric and validate the effectiveness of OCMOT through extensive evaluations using state-of-the-art methods.

---

> ### Author Rebuttal · Authors · 2024-08-17
>
> We appreciate Reviewer oy8h for acknowledging our work. We provide a detailed response to your questions in the following.
>
> >1. It would be nice to quantitatively show the reliability of the mgReA metric. For example, randomly select a number of results, run the CLIP matching approach and compare it with human annotators matching.
>
> We greatly appreciate your suggestion regarding the experiment on the reliability of mgReA.
> To ensure the reliability of our results, we randomly selected 40 videos from our dataset for comparison.
> Then we extracted the detection bounding boxes with an Intersection over Union (IoU) greater than 0.5 with the ground truth and subsequently annotated them manually, in which in total of 2,350 detection samples were obtained.
> It is noteworthy that, due to the LVIS dataset containing 1,203 categories, the workload for manual annotation was substantial.
> We check for the category matching using the following steps.
> If a subclass term existed, we selected it.
> Otherwise, if only a broader class term was present, we opted for the broader term.
> In cases without exact matches, we annotated from general to specific categories.
> In addition to annotation, we conducted thorough checks to ensure accuracy by someone else.
> After a few days of annotation by several graduate students, we averaged the results from the samplings.
>
> As shown in the below table, on the base class, the mgReA value is 26.65% for the CLIP matching method, and 25.05% for the manual annotation. On the novel class, the mgReA scores are 23.30% and 24.65%, for CLIP and annotation, respectively.
> We can see that, the differences between these results are minimal, with the mgReA of the base class differing by only 1.60% and for the novel class by 1.35%.
> This strongly supports the reliability and usefulness of the CLIP-based mgReA metric.
>
> |                        | Base mgReA | Novel mgReA |
> |------------------------|------------|-------------|
> | CLIP Matching          | $26.65\%$     | $23.30\%$      |
> | Annotators Matching | $25.05\%$ ($\Delta1.60$) | $24.65\%$ ($\Delta 1.35$) |
>
> Note that, the selected 2,350 manually annotated detection samples only account for approximately 1% of the total bounding boxes in our dataset. Considering the tremendous labor, although still with some small difference, the CLIP based matching used in this work is reasonable.
>
> >2. A minor issue: the template used by this paper is not the correct NeurIPS Dataset track review template.
>
> Thank you for your reminder. We will update the template in the final version.

---

> ### Comment · Area_Chair_ZEZR · 2024-09-01
> **Review/rebuttal discussion**
>
> Dear Reviewer oy8h,
>
> Even though your rating is positive, please reply to the authors rebuttal to let us know your opinion about the paper after reading it.
>
> Thanks

---

### Official Review · Reviewer_SS2z · 2024-07-25
**Review comments**

**Rating:** 4
**Confidence:** 5
**Correctness:** Yes.
**Clarity:** Yes.

**Review:**

1.	The proposed mgReA is based on TETA [13] with minor modifications. The multi-granularity is evaluated by WordNet but the calculation is not shown in this paper.
2.	The proposed OCTracker is to exploit the FlanT5-base [62] to obtain the capability of open corpus so it is not fair to compare it to open-vocabulary MOT such as OVTrack [15].
3.	The performance is worse than the OVTrack [15] in Table 1 so why do we need to use OCTracker?

**Strengths:**

Open-vocabulary MOT is an interesting idea and a new metric mgReA is proposed.

**Additional Feedback:**

No.

**Documentation:**

Yes.

**Ethics:**

No.

**Limitations:**

Yes.

**Opportunities For Improvement:**

1.	Show more comparisons and experiments to explain the advantages of open-vocabulary MOT.
2.	Provide the details of how to apply WordNet for multi-granularity evaluation.

**Relation To Prior Work:**

Yes.

**Summary And Contributions:**

They develop OCTrackB, a comprehensive benchmark for OCMOT, offering a balanced set of base and novel classes with corresponding samples. A multi-granularity recognition metric is proposed for better evaluation. Extensive evaluations of state-of-the-art methods demonstrate the validity of OCMOT and the advantages of OCTrackB.

---

> ### Author Rebuttal · Authors · 2024-08-17
>
> Thanks for your comments on this work. We provide a detailed response to address your concerns and clarify some misunderstandings.
>
> > 1. The multi-granularity is evaluated by WordNet but the calculation is not shown in this paper. Provide the details of how to apply WordNet for multi-granularity evaluation.
>
> We clarify the details of applying the WordNet for the multi-granularity evaluation.
> 1. **Hierarchical network construction of WordNet.**
> WordNet assigns unique IDs to nouns and maps their network relationships. We leverage WordNet to build a hierarchical network of category nouns, by annotating the tree with the category in the dataset as nodes.
> 2. **Hierarchical relation uncertainties.** Simply choosing the closest node to the root as a parent is ineffective in WordNet, because nodes can have multiple paths leading to them.
> For instance, as shown in Figure R1 (in the attached pdf file), the node "steak_knife.n.01" has two paths from "entity.n.01": a longer one (in orange) and a shorter one (in blue). We desire a steak knife to be a sub-class of knife.
> But opting for the shorter path might wrongly classify "steak_knife" as not a subclass of "knife".
> This means the hierarchical length can not directly reflect the subclass relations.
> 3. **The proposed greedy algorithm based solution.** We implement a greedy algorithm to resolve the above uncertainty problem by calculating the number of child nodes across aggregated paths for each LVIS category node. The goal is to maximize the total number of child nodes for selected parent nodes, effectively organizing related categories. After selecting a parent node, assign its children under it and repeat until all nodes are selected.
> 4. **Multi-granularity evaluation.** Based on the effective hierarchical relations, we design the multi-granularity metric. For a given category, we extend it up one hierarchy to a coarser granularity (e.g., from "shepherd dog" to "dog"), as shown in Figure 5 in the paper. This metric overcomes the limitations of the previous classification score ClsA.
>
> Therefore, we clarify that __although the calculating form of mgReA is similar to ClsA in TETA, the modification is nontrivial and rational__.
> For instance, in Table 1 of the paper, QDTrack and TETer both get a ClsA score of 0.1%, making the evaluation indistinguishable.
> MgReA can distinguish their performances clearly, with scores of 7.6% versus 2.5%.
>
> As per your suggestion, we will include the details of applying WordNet in our final version.
>
> >2. The proposed OCTracker is to exploit the FlanT5-base to obtain the capability of open corpus so it is not fair to compare it to open-vocabulary MOT such as OVTrack.
>
> First, it is essential to clarify that the proposed OCMOT (Open-Corpus) problem is distinct from the OVMOT (Open-Vocabulary) problem. Specifically, as shown in Figure 1 of the paper, OVMOT necessitates a predefined category list during inference.
> In contrast, OCMOT is category-free, eliminating the need for such a category list.
> Therefore, _OCMOT presents a greater challenge than the OVMOT problem due to the absence of a prior category list_.
> When comparing the OCTracker (for OCMOT) and the OVTrack (for OVMOT), *regarding the input conditions*, **this comparison favors the OVTrack over the OCTracker**.
>
> The proposed OCMOT problem handles the object recognition task as a generation problem, which is more general and practical than the classification problem in OVMOT.
> For this purpose, OCTracker uses a pre-trained language model FlanT5-base to generate the object category.
> OVTrack also uses the pre-trained multi-modal model CLIP for the classification problem.
> So, although *regarding the methodology*, it's also hard to compare the fairness.
>
> Overall, this work does not aim to compare with OVTrack for performance improvement under OVMOT setting.
> Instead, we introduce a new OCMOT problem (a further step of OVMOT) by establishing a benchmark and new method to lay the foundation for this research.
>
> >3. The performance is worse than the OVTrack in Table 1 so why do we need to use OCTracker?
>
> There may be a misunderstanding of the motivation of OCTracker.
> As discussed above, when comparing the results of OCTracker with OVTrack, the comparison favors the OVTrack over OCTracker, in terms of the input and problem difficulty.
> The reason we develop OCTracker is not its higher performance under previous OVMOT setting, but its ability to address the more practical problems of OCMOT.
>
> As stated by reviewer H893, “**Though we have seen tasks on open vocabulary (OV) MOT, _moving forward to an open-corpus (OC) one is a step forward_**." Also, as stated by reviewer oy8h, "This benchmark improves previous open-vocabulary MOT by requiring the model to suggest the categories, which is **a natural next step**."
>
> From a long-term perspective, AI is advancing toward more generalization. We foresee the MOT field also moving beyond predefined categories to open-corpus categories, with OCTracker pioneering this shift.
>
> >4. Show more comparisons and experiments to explain the advantages of open-vocabulary MOT.
>
> As clarified before, the purpose of this work is not to develop an open-vocabulary MOT method but to study a new problem of open-corpus MOT.
> This way, we are willing to provide an intuitive qualitative result to show _the advantages of OCMOT_.
>
> Specifically, we use an Internet video (involving a "chickadee") not included in the dataset, for testing, to investigate the effectiveness of different methods in real-world applications. Please see Figure R2 in the attached pdf file.
>
> 1. OVTrack, with the original setting, identifies only broad classes like "bird".
> 2. OVTrack, with the newly added category prompt, remains to provide an unstable result.
> 3. OCTracker, with only the video, delivers much more diverse outputs with multiple category levels and detailed descriptions.
>
> This highlights the advantage of OCMOT over OVMOT. For more details, please reference Figure R2's caption.

---

> > ### Comment · Reviewer_SS2z · 2024-09-01
> >
> > I appreciate the effort for the rebuttal from the authors. After reading the rebuttal, I still think the contribution is limited. As reviewer H893 mentioned, this is a simple combination of the existing methods. This work is very engineering. I intend to maintain my ratings. Maybe I have more expectations for NeurIPS but  I will respect the decisions of other reviewers.

---

> > ### Author Rebuttal · Authors · 2024-09-01
> >
> > The reviewer does not mention the concerns in the first-round review anymore, which we think have been fully addressed in the rebuttal. However, the reviewer raises a new question about the contribution, mainly focusing on the proposed method.
> >
> > Once again, as clarified in the introduction of the paper, the main contributions of this work are from three aspects: 1) A new problem of OCMOT, an extension of OVMOT with more practical application, 2) A new benchmark OCTrackB, a large-scale and comprehensive benchmark for the OCMOT problem, 3) The first baseline method for OCMOT and extensive comparison results with SOTA methods. The main purpose of this work is to build the foundation for OCTrack, with an available benchmark and a baseline method. So, **it is unfair to evaluate our work by only considering the method.**
> >
> > Although the method is based on some generally used models, the framework we proposed is highly effective and extensible, providing a solid foundation for future research. The proposed method, OCTracker breaks through the class-agnostic object detector, the language model based recognizer, and the association module, within a unified framework, which **makes open-corpus tracking grow out of nothing.** It is non-trivial and effective. Experimental results demonstrate that OCTracker achieves very promising results, i.e., comparable performance with the previous OVMOT methods requiring an additional category list during testing.
> >
> > Moreover, we do not agree that this work is engineering. It is essential to clarify that our submission pertains to the Datasets and Benchmarks Track, not the main track of NeurIPS. **This specific track aims to focus on the introduction of high-quality datasets and benchmarks, rather than merely presenting complex methods to address existing problems.**
> >
> > We understand the reviewer’s criteria for the classical NeurIPS, but we still wish you to consider the full contributions of this work and the expectations of the track to which we are submitting.

---

> ### Author Response · Authors · 2024-08-21
> **Follow-up on Discussion Section**
>
> Dear Reviewer SS2z,
>
> Thank you once again for your efforts in reviewing our work. We greatly appreciate your feedback and have addressed all the issues you raised. We are eager to discuss any further questions you might have.
>
> Best regards,
>
> Authors

---

> ### Author Response · Authors · 2024-08-25
> **Follow-up on Discussion Section (Again)**
>
> Dear Reviewer SS2z,
>
> First, thank you once again for your efforts in reviewing this work.
>
> As the discussion deadline approaches, we are eager to obtain your feedback on the response at your earliest convenience.
>
> We sincerely hope you to re-evaluate this work considering the clarification in the response. We are also willing to further discuss with you if there are still any concerns remaining.
>
> Best regards,
>
> Authors

---

### Official Review · Reviewer_H893 · 2024-07-30
**This paper proposed OCTrack, the first benchmark on open-corpus MOT.**

**Rating:** 8
**Confidence:** 4

**Review:**

This paper proposed a new task, Open-Corpus Multi-Object Tracking, together with a new dataset, a baseline method, and evaluation metrics. Though we have seen tasks on open vocabulary MOT, moving forward to an open-corpus one is a step forward. Overall I think this paper is of high quality and recommend acceptance.

**Strengths:**

This paper has clear paper writing. I appreciate the author's efforts on the list of 4 principles in benchmark construction. This makes paper reading clear and straightforward.

Significantly larger dataset size.

Extensive experiments on shows

A relative challenge benchmark. The performances of existing methods still are generally poor and have space for improvements.

**Additional Feedback:**

I am happy to raise my score if the author can address my concerns.

**Clarity:**

This paper is generally well-written.

In section 4, the author mentioned "We use a diffusion model to generate its adjoint image with the same object categories but different styles". I think this part lacks details references and details. How is this model trained? Which specific model is used?

**Correctness:**

It would be better if error bars were reported. For example, OCTracker only achieves 0.1% higher accuracy than the OC-based method DiffuTrack in meReA in the Base Class setting. It is needed to show such improvement is stable.

**Documentation:**

This dataset is provided in Google Drive and formatted well in COCO format.

**Ethics:**

This dataset is conducted using existing datasets. Thus there are no extra ethics concerns.

**Limitations:**

The new method OCTracker is a combination of many existing models. Thus the novelty is relatively limited.

**Opportunities For Improvement:**

How does OCMOT give labels for images from TAO and LV-VIS? I think it is a critical part of this paper but not discussed in detail.

**Relation To Prior Work:**

This benchmark directly uses images from existing work TAO and LV-VIS, with reformation and new open-corpus labels.

**Summary And Contributions:**

This paper proposed a new task, Open-Corpus Multi-Object Tracking, together with a new dataset, a baseline method, and a evaluation metrics.

---

> ### Author Rebuttal · Authors · 2024-08-17
>
> We appreciate Reviewer H893 for acknowledging our work and being willing to raise the score by considering the response.
>
> >1. How does OCMOT give labels for images from TAO and LV-VIS? I think it is a critical part of this paper but not discussed in detail.
>
> Thanks for your reminder.
> The proposed dataset for OCMOT contains three types of labels for each object, i.e., the object bounding box, object category name, and track ID.
>
> 1. For the bounding box annotation, TAO provides the box annotations that can be directly used. For LV-VIS, a segmentation dataset, we generate the box from the segmentation boundary by using its maximum values at four sides.
>
> 2. For the category name, we can directly obtain the object category from each annotated bounding box in TAO or each segmentation boundary in LV-VIS.
>
> 3. For the track ID, TAO provides the ID as a tracking dataset. For LV-VIS, we obtain the continuous ID of an object by matching the target with the maximum IOU (Intersection over Union) in the next frame, where the IOU is calculated by the ground-truth segmentation annotation.
>
> We will include this part in the final version of our paper.
>
> >2. The new method OCTracker is a combination of many existing models. Thus the novelty is relatively limited.
>
> First, although the new method OCTracker in this work combines some existing models, it is the __first available framework for OCTrack problem__. The proposed method can be used to provide the basic baseline for this promising problem.
>
> Next, we clarify that the main technical contribution and novelty of OCTracker lie in that it __makes open-corpus tracking feasible__, especially the object recognition task from the classification problem into the generation problem possible.
> This is achieved by the OCTracker that _breaks through_ the class-agnostic object detector, the language model based recognizer, and the association module, _in a unified framework_.
> OCTracker also achieves very promising results, i.e., comparable performance with the previous OVMOT methods requiring an additional category list during testing.
>
> Last, the main purpose of this work is to build the foundation for OCTrack, with an available benchmark, and a vanilla and effective baseline method. So the method does not involve many complex technologies.
> We hope the methods for such a problem can be gradually improved through our efforts along with contributions from the community in the future.
>
> >3. It would be better if error bars were reported. For example, OCTracker only achieves 0.1% higher accuracy than the OC-based method DiffuTrack in meReA in the Base Class setting. It is needed to show such improvement is stable.
>
> As per your suggestion, we randomly sampled 30% of the videos for each class from the dataset and repeated this process ten times.
> For each indicator, we calculated the mean and standard error of performance metrics across iterations.
> These values were used to generate the Standard Error of the Mean (SEM) error bars, providing insight into the variability and reliability of the "OCTracker" and "GenerateU+DiffuTrack" methods across different dataset samplings. The results are shown in Figure R1 in the attached pdf file.
>
> From the fifth pair of bars, we can see that, for the comparison of OCTracker and the OC-based method (GenerateU + DiffuTrack) under the meReA metric as mentioned by the reviewer, the mean performance of our method is only slightly higher than that of the comparison method, the pronounced difference in the error bars illustrates its performance advantage.
> Additionally, the shorter length of the error bars in our method indicates a lower degree of variability or uncertainty.
> We can also see from all the results that, our OCTracker achieves consistently stable improvements across various metrics compared with the comparison method "GenerateU + DiffuTrack".
>
> >4. In section 4, the author mentioned "We use a diffusion model to generate its adjoint image with the same object categories but different styles". I think this part lacks references and details. How is this model trained? Which specific model is used?
>
> As mentioned in the paper, the data hallucination strategy for generation follows previous work [15].
> This method is based on a denoising diffusion probabilistic model (DDPM).
> Specifically, given an original image and its object detection annotations, we generate a transformed image (e.g., with rotation, scaling).
> We input the transformed image into the diffusion model to progressively denoise it.
> During this, the foreground regions are kept fixed at each iteration.
> After that, we obtain the adjoint image of the original image with the same object categories but different styles.
> For the implementation of the diffusion model, we use the pre-trained "Stable Diffusion v1-4" model [A], which integrates a "CLIP ViT-L/14" text encoder with an autoencoder and a latent diffusion model. The pre-training is implemented on the "laion-aesthetics v2 5+" dataset [B].
>
> We will include this part in the supplementary material of our final paper.
>
> ### References
> [15] Li, Siyuan, et al. "OVTrack: Open-vocabulary multiple object tracking." In Proceedings of the IEEE/CVF conference on computer vision and pattern recognition, 2023.
>
> [A] Rombach, Robin, et al. High-resolution image synthesis with latent diffusion models. In Proceedings of the IEEE/CVF conference on computer vision and pattern recognition, 2022.
>
> [B] Schuhmann, Christoph, et al. Laion-5b: An open large-scale dataset for training next generation image-text models. Advances in Neural Information Processing Systems, 2022.

---

> > ### Comment · Reviewer_H893 · 2024-09-01
> > **Response**
> >
> > Thank you for your detailed response. My concerns have been satisfactorily addressed. However, I concur with Reviewer SS2z that the novelty of OCTracker appears somewhat limited. While I appreciate the introduction of a new benchmark and plan to give this paper a high rating for that contribution, I am hesitant to award it higher marks due to the aforementioned limitations.

---

> > > ### Author Response · Authors · 2024-09-01
> > >
> > > We are very pleased to hear that your concerns have been satisfactorily addressed. Additionally, we sincerely appreciate your recognition of our new dataset and benchmark. We hope that in the future, more researchers will engage with this novel task and contribute even more effective methods.

---

> ### Author Response · Authors · 2024-08-22
> **Follow-up on Discussion Section**
>
> Dear Reviewer H893,
>
> Thank you for your positive feedback on our work. We greatly value your insights and have carefully addressed the concerns you raised. We look forward to your appreciated guidance in further discussions on our responses.
>
> Best regards,
>
> Authors

---

> ### Author Response · Authors · 2024-08-29
> **Follow-up on Addressing Your Concerns**
>
> Dear Reviewer H893,
>
> We hope this message finds you well. With the discussion deadline approaching, we are eager to know if our responses have effectively addressed your concerns. If they have, we would greatly appreciate an improvement in the rating of our work.
>
> We look forward to your response.
>
> Best regards,
>
> Authors

---

### Official Review · Reviewer_gHx8 · 2024-08-09
**Large open-corpus multi-object tracking dataset**

**Rating:** 7
**Confidence:** 2
**Correctness:** Claims look correct.
**Clarity:** Paper is clearly written.

**Review:**

The open-corpus multi-object tracking (OCMOT) is an interesting problem and given the proliferation of vision-language models the problem of OCMOT could be useful in video understanding of VLMs.

**Strengths:**

1. Authors introduce open-corpus multi-object tracking (OCMOT) problem and release a new dataset, OCTrackB towards that end.
2. They propose an new evaluation metric and baseline method for OCMOT.

**Additional Feedback:**

See comments.

-----

The authors have addressed my 2nd concern and showed that it is not simple to create their proposed dataset using simple scripts. The choice of balanced dataset seems more subjective than objective. Yes, authors have 2 references stating the benefit of balanced dataset but at the same time there are many more references that can be shared to handle skewed data. Having said that, the proposed dataset is a good starting point for researchers where they don't have to deal with an additional problem (of skewed data) along with OCMOT.  Hence, I am increasing the paper rating.

**Documentation:**

Dataset is available.

**Ethics:**

No.

**Limitations:**

Limitations are discussed in supplementary.

**Opportunities For Improvement:**

1. Many real-world scenarios have skewed data. It is not why authors prefer balanced dataset. Moreover it is very difficult to achieve balance among classes in multi-object setting. The balance constraint imposed by authors would have result in some filtering of videos which would have unnecessary.
2. The work relies on existing tracking dataset and filtering criteria to create new dataset. The whole process is automated and at least from the no manual verification is performed in the dataset creation process.

**Relation To Prior Work:**

yes.

**Summary And Contributions:**

The authors release a large open-corpus, diverse, balanced multi-object tracking dataset with multiple base and novel classes. They also propose an multi-granularity recognition metric along with a baseline method for open-corpus multi-object tracking. Authors evaluate existing multi-object tracking methods on their dataset OCTrackB, and show that difficulty of proposed novel dataset.

---

> ### Author Rebuttal · Authors · 2024-08-17
>
> Thank you for pointing out that the proposed problem is interesting and useful and acknowledging the significance of this work. We provide a detailed response to your questions in the following.
> > 1. Many real-world scenarios have skewed data. It is not clear why authors prefer a balanced dataset. Moreover, it is very difficult to achieve balance among classes in a multi-object setting. The balance constraint imposed by the authors would have resulted in some filtering of videos which may have been unnecessary.
>
> Thanks for your constructive question.
>
> First, we acknowledge that real-world data is often skewed and imbalanced. However, as mentioned in Section 3.4 of the paper, as an evaluation benchmark, we try to maintain category balance to **ensure that abundant but simple classes do not dominate the evaluation**. Here we provide detailed clarification of this point. Many real-world scenarios often involve skewed data. However, this skewness leads to significant evaluation issues due to class imbalance. Specifically, as discussed in previous works [A] and [B], imbalanced datasets present two main problems:
> 1. **Distortion of evaluation metrics.**
> Class imbalance can easily distort the performance metrics, like accuracy, precision, and F1 score, in which accuracy is the most commonly used metric in OVMOT/OCMOT. This can lead to biased evaluation results, potentially producing unrealistically optimistic or pessimistic results. For example, the results may appear falsely optimistic when samples from easier classes dominate the dataset.
> 2. **Misleading of comparison results.**
> Imbalanced datasets can lead to majority classes overshadowing minority classes, which creates an illusion of strong or poor overall performance. This distortion can significantly mislead average evaluation metrics in OVMOT/OCMOT. Consequently, comparisons among the methods can become unreliable.
>
> Furthermore, many classical multi-object tracking (MOT) tasks mainly focus on specific classes, such as persons and vehicles. So we hypothesize that the tracking performance on the above common classes has reached a relatively good and stable level.
> Differently, this work introduces a pioneering benchmark for open-class tracking. Therefore, we designed OCTrackB as a class-balanced dataset to ensure that **evaluations consider a broader range of categories and are not dominated by the common (easy) classes with the majority of samples**.
>
> Finally, we also contend that the proposed OCTrackB dataset and the existing TAO-val/test dataset (unbalanced classes) can be complementary in jointly promoting open-class tracking.
>
> > 2. The work relies on existing tracking datasets and filtering criteria to create a new dataset. The whole process is automated and at least, no manual verification is performed in the dataset creation process.
>
> **Guarantee of the basic datasets.** Similar to previous open-class works using the existing datasets to build new benchmarks (e.g., the first open-vocabulary tracking OVTrack [C] and the first open-world tracking OWTrack [D] both directly use the TAO dataset for evaluation), the proposed benchmark is built based on two existing datasets, i.e., LV-VIS and TAO, whose **annotation accuracy is verified and guaranteed**.
> Specifically, the LV-VIS dataset ensures quality by accurately annotating visible objects with categories and implementing cross-revision for quality control. The TAO dataset guarantees quality by the hierarchical annotation for precision, where the re-annotation process confirms a high IoU for accuracy. TAO is further maintained through automation, manual review, and statistical analysis.
>
> **Purpose and contribution of this benchmark.** In this work, unlike the direct usage of existing datasets [C][D], we design a series of principles, including the standardness principle, category enrichment principle, sample enrichment principle, and semantic compatibility principle, as discussed in the paper. To abide by these principles, although based on existing data (video and annotation), the new benchmark OCTrackB is more appropriate and useable for the new task of OCTrack.
>
> **Data integration and manual verification.** Our workload focuses on data screening, integration, and unification, e.g., filtering the videos using a greedy algorithm, and unifying the annotations into the COCO format.
> Additionally, it is worth mentioning that we also _manually unify the category definitions across different datasets_.
> For example, multiple vocabularies may have the same semantic meaning (e.g., the “police car” in LVIS and "police cruiser" in LV-VIS), one vocabulary with multiple meanings (e.g., the "bat" can refer to both an animal and a piece of sports equipment), and the same object may represent the vocabularies of different granularities (e.g., the "dog" category includes breeds such as "bulldog", "dalmatian", "pug" and "shepherd dog").
> To unify the category definition, we manually reviewed all categories in the original datasets, and unified them into the 1,203 categories defined in the LVIS.
> This verification process involves manual annotation and double-checking to ensure uniformity.
> Considering the proposed dataset with numerous videos and categories, this manual verification is labor-intensive.
>
> ### References
> [A] Luque, Amalia, et al. The impact of class imbalance in classification performance metrics based on the binary confusion matrix. *Pattern Recognition*, 2019.
>
> [B] Jeni, László A., et al. Facing imbalanced data -- recommendations for the use of performance metrics. *Humaine Association Conference on Affective Computing and Intelligent Interaction*, 2013.
>
> [C] Siyuan Li, et al. Ovtrack: Open-vocabulary multiple object tracking. In Proceedings of the IEEE/CVF Conference on Computer Vision and Pattern Recognition, 2023.
>
> [D] Yang Liu, et al. Opening up open world tracking. In Proceedings of the IEEE/CVF Conference on Computer Vision and Pattern Recognition, 2022.

---

> ### Comment · Area_Chair_ZEZR · 2024-09-01
> **Rebuttal/review discussion**
>
> Dear Reviewer gHx8,
>
> Even though your rating is positive, please reply to the authors rebuttal to let us know your opinion about the paper after reading it.
>
> Thanks

---

### Decision · Program_Chairs · 2024-09-26

**Decision:**

Reject

**Comment:**

This paper proposes a new benchmark, OC-MOT, for Open-Corpus MOT (Multi-Object Tracking). The paper presents a new formulation of the OC-MOT, the Dataset Collection and Annotation (using TAO and LV-VIS) pipeline, new metrics, a baseline and experimental results of the baseline in the new dataset. Most reviewers favor the paper thanks to its new task setup and thorough experiments. However, the remaining concerns include:
  * OC-MOT is an incremental step from OV-MOT (e.g., "open vocabulary MOT, moving forward to an open-corpus one is a step forward"). We understand the difference between OC-MOT and OV-MOT, but the question here is if the one-step-moving-forward from OV to OC is significant enough for a top-tier conference paper? Since OV-MOT is already a sub-problem in visual tracking, thus OC-MOT is just providing another evaluation setup of MOT, e.g., evaluated in the Closed-Vocabulary, OV, or OC setting.
  * The novelty of the dataset is limited as it is composed from two existing datasets, which already provided annotations. The contribution of this paper is to perform sub-sampling (greedy algorithms from TAO and LV-VIS), to split base / novel classes, and to provide baseline results. As a result, the contribution of this paper can also be considered as providing a new evaluation setup for MOT (aka OC-MOT), evaluating baselines on the newly-remixed dataset, OC-MOT (from TAO and LV-VIS).
  * It is unclear about the usefulness of the new problem setup, e.g., from OV-MOT to OC-MOT. Why do we need a OC-MOT dataset? It would be great if the paper can demonstrate the essence of OC-MOT approaches against the current OV-MOT, or even standard MOT, or a simple combination of any class-agnostic segmentation / tracking methods (e.g., SAM, flow-SAM) + LLM-based caption-generation? The paper misses various important baselines to backup the claim that OC-MOT is necessary.
  * There are potential biases in video sampling from TAO and LV-VIS (long-tail natural distribution vs. fully class-balanced sampling).

Because of the above concerns, AC recommends not accept the paper in its current form and encourages the authors to improve their work based on the above suggestions (e.g., the novelty / importance of OC-MOT vs. OV-MOT, and conduct more baselines / experiments to demonstrate the of OC-MOT task / dataset is essential) and re-submit to future conferences.